# Global-scale magnetosphere convection driven by dayside magnetic reconnection

Lei Dai [1] ✉, Minghui Zhu [1], Yong Ren[1], Walter Gonzalez[1,2], Chi Wang [1], David Sibeck[3], Andrey Samsonov[4], Philippe Escoubet[5], Binbin Tang [1], Jiaojiao Zhang[1] & Graziella Branduardi-Raymont [4,6]

Plasma convection on a global scale is a fundamental feature of planetary magnetosphere. The Dungey cycle explains that steady-state convection within the closed part of the magnetosphere relies on magnetic reconnection in the nightside magnetospheric tail. Nevertheless, time-dependent models of the Dungey cycle suggest an alternative scenario where magnetospheric convection can be solely driven by dayside magnetic reconnection. In this study, we provide direct evidence supporting the scenario of dayside-driven magnetosphere convection. The driving process is closely connected to the evolution of Region 1 and Region 2 field-aligned currents. Our global simulations demonstrate that intensified magnetospheric convection and field-aligned currents progress from the dayside to the nightside within 10–20 minutes, following a southward turning of the interplanetary magnetic field. Observational data within this short timescale also reveal enhancements in both magnetosphere convection and the ionosphere's two-cell convection. These findings provide insights into the mechanisms driving planetary magnetosphere convection, with implications for the upcoming Solar-Wind-Magnetosphere-Ionosphere Link Explorer (SMILE) mission.

One fundamental and prominent feature of planetary magnetosphere is plasma convection induced by surrounding solar wind[1–6]. The Dungey cycle[7], a global-scale pattern of magnetosphere convection, currently serves as the basis for most of our thinking about solar-wind-magnetosphere coupling[8,9]. The dynamics of magnetosphere convection are crucial in the occurrence of geomagnetic storms and substorms[10–14].

In the Dungey cycle, magnetic-reconnection-driven convection plays a central role[7]. A schematic of the sequential process of enhanced convection is shown in Fig. 1a. At the dayside boundary of the magnetosphere, the interaction of a southward interplanetary magnetic field IMF with the northward magnetic field results in magnetic reconnection. This dayside reconnection drives plasma convection of open-field lines[7,15–17], flowing anti-sunward over the geomagnetic poles

to the nightside. Over a timescale of approximately one hour, the accumulation of plasma and associated magnetic flux may trigger magnetic reconnection in the nightside magnetospheric tail, driving sunward plasma convection[11]. The nightside reconnection is responsible for the convection in the closed part of the magnetosphere, where magnetic field lines connect to the Earth's surface. The sunward convection from nightside reconnection eventually deflects azimuthally in the tail-to-dipole transition region of the magnetosphere, taking the form of a convective flow deflected around Earth[1,7]. Correspondingly, the Dungey convection in the ionosphere is roughly a two-cell convection pattern, characterized by antisunward flow across the polar cap and sunward flow at lower latitudes[7].

The Dungey cycle provides a steady-state depiction of magnetosphere convection[7,18]. Time-dependent convection is explored by the

[1]National Space Science Center, Chinese Academy of Sciences, Beijing 100190, China. [2]National Institute for Space Research (INPE), São José dos Campos, São Paulo, Brazil. [3]Goddard Space Flight Center, NASA, Greenbelt, US. [4]Mullard Space Science Laboratory, University College London, Dorking, UK. [5]European Space Research and Technology Centre, European Space Agency (ESA), Noordwijk, Netherlands. [6]Deceased: Graziella Branduardi-Raymont.
✉ e-mail: ldai@spaceweather.ac.cn

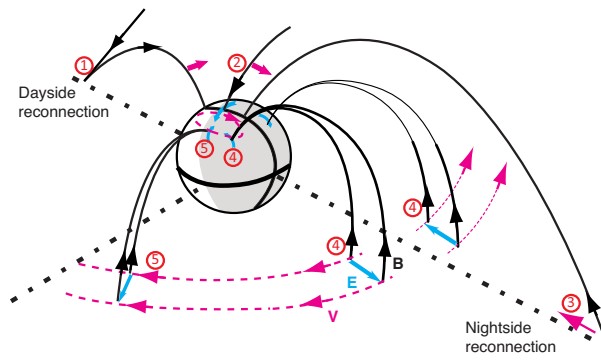

a) Dungey-Cycle Convection

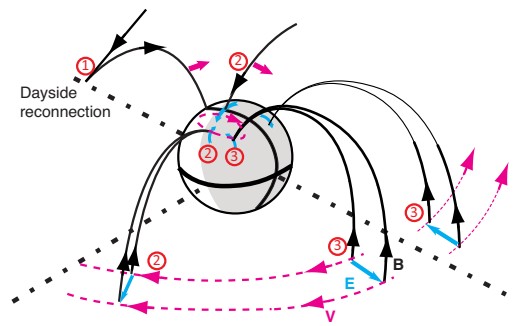

b) Dayside-Driven Convection

**Fig. 1 | A schematic of the sequence of enhanced magnetosphere-ionosphere convection in the Dungey Cycle (nightside-driven convection) and the Dayside-Driven convection. a** The nightside-driven convection, and **b** dayside-driven convection are independent[14,19,20]. black arrows: magnetic field lines; pink arrows: convective flow; blue arrows: convection electric fields.

expanding/contracting polar cap (ECPC) model of the Dungey cycle[19–21], showing diagrams of convection flows induced by purely dayside or night reconnection[20,22,23]. The relative strength of the dayside and nightside driven convection determines the cross-polar-cap electric potential[18,24]. Given the length and strength of the day-side reconnection, the ECPC model can calculate the ionosphere convection pattern induced only by dayside reconnection[19,21]. Figure 1b illustrates the expected magnetospheric convection driven solely by dayside reconnection. The chain of dayside-driven convection is expected to occur within a short timescale (10-20 minutes) of re-establishing the ionosphere's two-cell convection[14,20]. This timescale reflects the rapid phase propagation of a state of enhanced convection across the global magnetosphere and ionosphere. Within this 10-20 minute window, the plasma associated with a specific field line moves only a relatively short distance throughout the entire convection cycle, which typically lasts 2-4 hours[11].

From a theoretical standpoint, it is justifiable to consider dayside reconnection as the only driver of magnetospheric convection. The effect of nightside reconnection and the associated sunward convection can theoretically be considered as a boundary condition for convection electric fields and plasma at 10–15 Earth radii ($R_E$) on the nightside[25]. Even when reconnection and sunward convection are disregarded at the nightside boundary, magnetospheric convection can proceed[20].

From an observational perspective, the concept of dayside-driven convection is supported by ionospheric measurements. The reconfiguration of the ionosphere's two-cell convection in response to a southward IMF occurs rapidly in less than 10 minutes[11,26–28] and progresses from the dayside to the nightside[29–31]. These ionospheric

observations imply a quick re-establishment of a dayside-driven magnetosphere convection, given an anticipated coupling between the ionosphere's convection and magnetosphere convection. This coupling is primarily provided by Region 1 and Region 2 Birkeland field-aligned currents (FAC)[32–34]. Region 1 FAC, flowing into the ionosphere at dawn and out of the ionosphere at the dusk, maps to the vicinity of the magnetopause. Region 2 FAC is adjacent to the Region 1 FAC and maps to the inner part of magnetosphere, with its direction opposite to that of Region 1 FAC.

In this study we utilize global simulations and observations to investigate dayside-driven magnetosphere convection. While previous studies of ECPC model capture essential features of the dayside-driven convection[18–21], our approach relies on different types of data and offers distinct advantages.

Firstly, previous ECPC model studies often focus on ionospheric convection data. Our approach involves the examination of global simulation and observational data of magnetospheric convection. The combination of global simulations with magnetospheric observations can provide direct evidence of dayside-driven magnetospheric convection. Secondly, studies involving simplified ECPC models typically specify the dayside reconnection driver to compute ionospheric convection for mathematical simplicity[19,21]. In contrast, our global simulations analyze the temporal evolution and the self-consistent interplay of magnetic reconnection, magnetospheric convection, and ionospheric convection. For example, the ionospheric convection electric fields and magnetosphere FAC resulting from dayside reconnection are self-consistently computed in a loop[35].

Our study is effectively a numerical version of ECPC model that directly address dayside-driven magnetospheric convection with actual data. Based on global simulations for a case study in a strong solar wind, we demonstrate that the enhancement of magnetospheric convection and associated Region 1/2 FAC progresses from the day-side to the nightside. The enhancement of magnetosphere-ionosphere convection responds rapidly within 10–20 minutes to the southward turning of IMF Bz, as evidenced by both simulations and observations.

## Results

### Dayside-driven convection in global simulations

In this study, we conduct global MHD simulations to investigate the response of the magnetosphere-ionosphere to a southward turning of the IMF. The solar wind condition is displayed in Fig. 2a. This turning occurs after a 2.5-hour period of northward IMF on March 11, 2016. We examine the Region 1 and Region 2 FAC on the ionosphere and the magnetosphere convection velocity (Fig. 2b1–b4) at 3-9 minutes after the southward turning ($T = 0$) of IMF. Figure 2b1–b4 display continuous snapshots of the FAC distribution in invariant latitude and magnetic local time (MLT). The Region 1 FAC, mainly between 70°–80° invariant latitude, is more intense than the adjacent Region 2 FAC between 60°–70° invariant latitude. Both Region 1-FAC and Region 2 FAC are intensified in $T = 3$–9 min. The progression of the Region 2 FAC and the dawn part of Region 1 FAC towards the nightside is particularly clear (Fig. 2b3, b4). Similarly, the magnetosphere convection velocity ($V_\phi$) on the GSM XY-plane first intensifies on the dayside and then extends to the nightside during $T = 3$–9 min (Fig. 2c1–c4). The direction of $V_\phi$ is counter-clockwise on the dawnside and clockwise on the duskside, consistent with mapping from the ionosphere's two-cell convection. The main contribution to $V_\phi$ is from the $V_x$ component, as shown in (Fig. 2d1–d4). $V_y$ also contributes to the convection pattern, particularly between 65°–75° at $T = 7$–9 min as evidenced in the supplementary Fig. S1.

The East-West Keogram (ewogram) in Fig. 3 further demonstrates a close relation between Region 1/2 FAC and the magnetosphere's convection. The ewogram shows a 2D MLT-time map of a quantity averaged within a specific range of invariant latitudes. In the 71°–74° latitude range (Fig. 3a1–b1), both Region 1 FAC and $V_\phi$ show

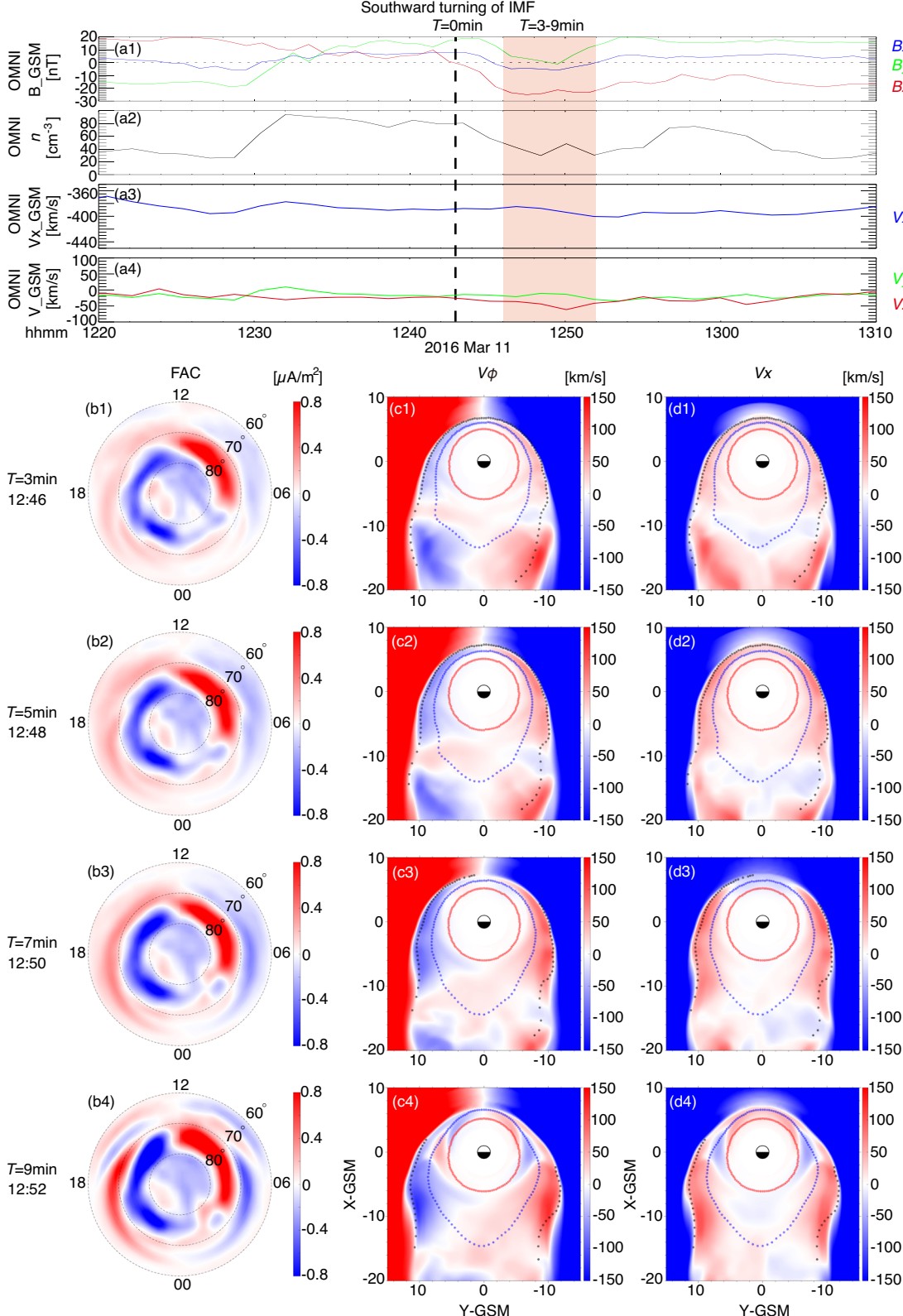

**Fig. 2 | Global MHD simulations of the magnetosphere-ionosphere response following a southward turning of IMF. a** OMNI data as the input to the global MHD simulation. Magnetic fields (a1) and plasma parameters (a2,a3,a4) in GSM as obtained from Wind spacecraft and time-shifted to the Earth's bow shock (-10$R_E$). **b1**–**b4**): Temporal evolution of the FAC at the top of the northern ionosphere. Positive sign (red) is the direction into the ionosphere. The magnetosphere convection velocity $V_\phi$ in the azimuth direction (**c1**–**c4**) and $V_x$ (**d1**–**d4**) at the XY-GSM plane. A positive sign of $V_\phi$ corresponds to counter-clock (eastward) direction. The dash lines of circles maps to the invariant latitude of 65°, 70°, and 75° in the radially outward direction.

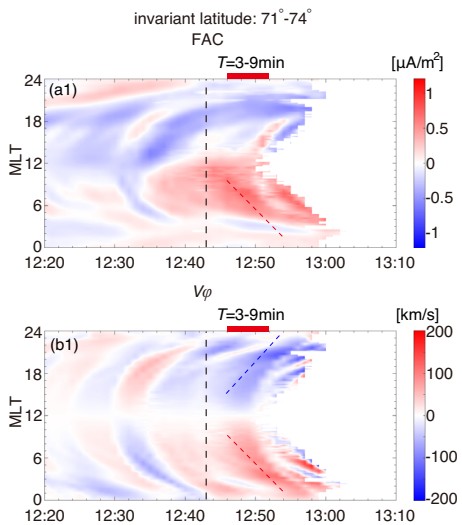

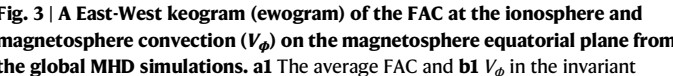

**Fig. 3 | A East-West keogram (ewogram) of the FAC at the ionosphere and magnetosphere convection ($V_\phi$) on the magnetosphere equatorial plane from the global MHD simulations. a1** The average FAC and **b1** $V_\phi$ in the invariant latitude 71°-74°. **a2** The average FAC and **b2** $V_\phi$ in the invariant latitude 65°-70°. Nan values are outside the closed-part of the magnetosphere. The Blue and red dash lines illustrate evolution along the east-west direction.

intensification following $T = 0$. The enhancement spreads from the dayside towards the nightside, exhibiting a clear positive slope on the duskside and a negative slope on the dawnside (indicated by red/blue dashed lines). This is consistent with the observed feature of Region 1 FAC shifting toward nightside during active geomagnetic condition[34]. The spreading of the intensification corresponds to roughly~1.0MLT/min. Before $T = 10$min, Region 2 FAC and the associated $V_\phi$ mostly occupy 65°-70 ° (Fig. 3a2-b2). During this interval, magnetosphere convection associated with Region-2 FAC is weaker than that associated with Region 1 FAC (Fig. 3b1, b2). Occasionally, Region 2 FAC extends to latitudes above 70° at MLT = 12-18 in Fig. 3a1 (also evident in Fig. 2b1-b4), which may affect the FAC pattern in >70°. After $T = 10$ min, Region 1 FAC and the associated $V_\phi$ move into the lower latitude range of 65°-70 ° (right blue/red dash lines in Fig. 3a2, b2). This movement is mainly due to a magnetic field erosion induced by continuous dayside reconnection.

Before the southward turning of IMF ($T = 0$), Region 1 FAC and magnetosphere convection are less intense but do not vanish (Fig. 3a1, b1). This is expected since Region 1 FAC has a permanent component even during quiet geomagnetic conditions[32]. During northward IMF, generation of Region 1 FAC and the convection pattern may be complicated by viscous-type interaction[36] and/or high-latitude lobe reconnection[18,28].

Both Figs. 2 and 3 consistently show that the dominant pattern of magnetosphere convection is from the dayside to the nightside at a radial distance <$10R_E$ within 30 minutes after the southward turning of the IMF. This convection pattern is difficult to be explained with nightside reconnection. Additionally, simulations do not show sunward plasma flow (Fig. 2d1-d4) or its associated flow deflection (Fig. 2c1-c4) coming from nightside tail reconnection during this interval. The dayside-driven convection in this event appears to be a general feature in our MHD simulations of southward IMF turning. Similar patterns have also been observed in other MHD algorithms simulations, such as the Lyon-Fedder-Mobarry (LFM)[37] global MHD model[38].

### Dayside-driven convection in observations

In Fig. 4, we present magnetosphere-ionosphere observations related to the dayside-driven convection event on March 11. Panels a-e display the temporal evolution of the PC-N index, AU/AL index, cross-polar-cap potential, magnetosphere convection and the ion energy

spectrum from Magnetospheric Multiscale (MMS) spacecraft. The red shaded box marks the first interval interval ($T = 1-9$ min) of enhanced convection, as analyzed in simulations in Figs. 2 and 3. The blue shaded box marks the interval of the second convection enhancement. Panels f-k show the invariant latitude-MLT map of ionospheric convection (colored flagpoles) and the electrostatic potential (contours) in these intervals. In our observations, we focus on two key aspects.

The first aspect is the prompt initiation of the ionosphere's two-cell convection following the southward turning of the IMF. Within $T = 1-9$ minute, we observe a rapid of the ionosphere's two-cell convection, as evidenced by increases in the PC-N index, the AU/AL index, and the polar cross-cap potential. The PC-N index and polar cross-cap potential indicate the strength of high-latitude ionosphere convection. AU index is a rough measure of the DP-2 current (two-cell convection) in the aurora latitude[12]. The increase of AU and AL is comparable in $T = 1-9$ min, indicating a enhanced two-cell convection (DP-2) without a substorm current system (DP-1)[14]. The first four snapshots of the ionosphere convection map (Fig. 4f-i) also depict a rise in convection (approximately $500 - 1000$ m/s) at latitudes > 65°. Notably, the most substantial enhancement (1000 km/s,yellow) progresses from MLT = 8-9 towards MLT = 7 from 12:48 UT to 12:52 UT in Fig. 4h, i.

The second aspect focuses on the coupling of ionosphere convection to magnetosphere convection. While comprehensive global-scale observations of magnetosphere convection are lacking[39,40], data from the MMS provide partial supporting evidence. With inherent uncertainties in the mapping in simulations, the location of MMS roughly maps to MLT~4.1 and an invariant latitude ~66°-72° after $T = 0$ in this event. During the first ($T = 3-9$min) and second ($T = 19-23$ min) enhancement, MMS observes a significant enhancement of convection consistent with a sunward return flow of ionosphere's two-cell convection (Fig. 4f-i).

Between 12:52 UT-12:56 UT, the magnetosphere convection appears to reverse, as indicated by MHD simulations (Fig. 3b1) and MMS observations (Fig. 4d). This reversal of magnetosphere convection is likely due to large-scale vortexes near the flank, accompanied by the formation of an additional ionosphere convection cell at aurora latitudes (See supplementary Fig. S2).

### Connection of enhanced convection to substorm

The second enhancement of convection occurs during $T = 19-23$ min (marked by the blue box), coinciding with a substorm expansion

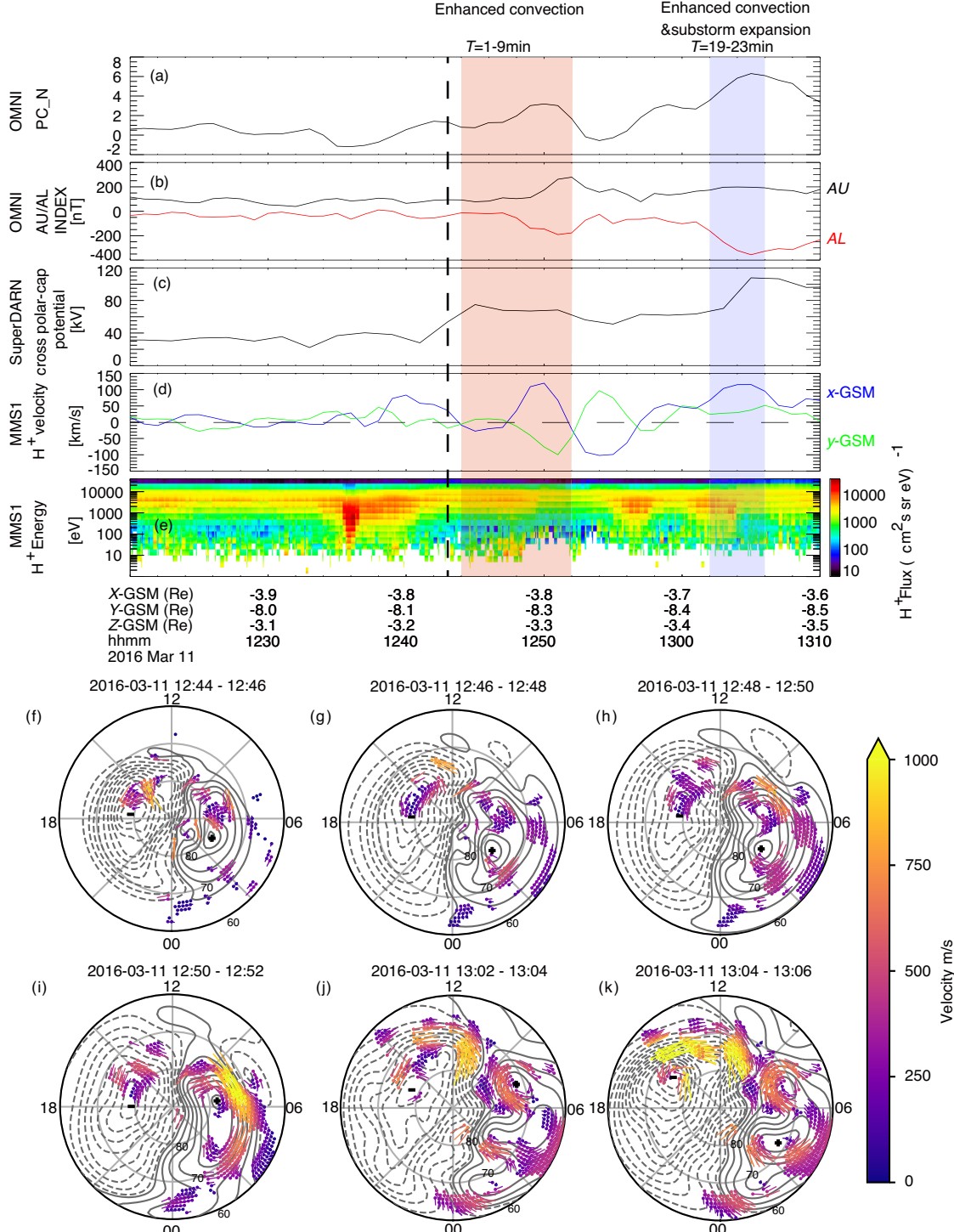

**Fig. 4 | Observations from ionosphere and magnetosphere on the Mar 11 event. a** Polar cap index, **b** AU and AL index, **c** the cross-polar-cap potential from SuperDARN, **d** Plasma convection velocity ($V_x$ and $V_y$) from HPCA instrument on MMS1, **e** Proton energy spectrum from MMS1, **f**–**k** Ionospheric convection maps.

denoted by a sharp decrease in the AL index. Correspondingly, the ionosphere convection map (Fig. 4j, k) reveals a enhancement (1000 km/s) of convection velocity at latitude > 60°. The weaker enhancement of 500–600 m/s flows at midnight are indicative of substorm expansion phase flows. During this second enhancement, SCW-type DP-1 FAC appears at MLT = 22–24 and MLT = 0–2 near 13:00UT at 65°–70° in global simulations (Fig. 3a2), supporting the implication that enhanced convection may be related to substorm expansion.

## Discussion

### Mechanism of dayside-driven convection

Our global simulations, as depicted Figs. 2 and 3, confirm a close correlation between Region 1/2 FAC and the dayside-driven magnetosphere convection. The following is a simple picture of the driving process. Initially, reconnected field lines from the dayside are dragged by the anti-sunward convection of the solar wind. These dragged field lines wrap around the magnetopause, forming a magnetic shear with the closed-field lines of the magnetosphere. Near the dawn and

dusk sector, the primary contribution to this magnetic shear is from the $\Delta B_x/\Delta Y$ component. This magnetic shear introduces Region 1 FAC. Subsequently, Region 1 FAC may lead to magnetic pressure reductions on the dayside, inducing sunward convection within the magnetosphere[11,18,41]. The dayside-driven magnetosphere convection then initiates Region 2 FAC and a portion of Region 1 FAC within the magnetosphere. This study primarily focuses on the direct evidence of dayside-driven convection, leaving a detailed analysis of the force balance, magnetic shear, and convection electric fields associated with this process for future investigations.

### Implications for substorms
The dayside-driven convection may have implications for substorms[14]. The driven magnetosphere convection corresponds to ionosphere's two-cell convection and its associated Hall current (DP-2). As the driven-convection progresses to the nightside, it produces a flow deflection in the azimuthal direction around Earth. These flow deflections may contribute to the formation of the substorm current system (SWC/DP-1)[12], as partially evidenced by our simulations near 13:00UT. In simple terms, dayside reconnection drives DP-2, which subsequently causes DP-1. However, it's important to note that the dayside-driven two-cell convection (DP-2) doesn't always correspond one-to-one with substorm expansion (DP-1). Enhanced magnetosphere convection can occur without subsequent substorm expansion[3,6].

### Role of the ionosphere in Magnetosphere Convection
In ECPC model and the global simulations, the ionosphere electric fields can be mapped out to influence magnetosphere convection. In this sense, the ionosphere might, to some extent, control magnetosphere convection through the ionospheric conductance[42,43]. In a test run of global simulation for the same event (supplementary Fig. S3), we increase the ionospheric conductance to $10^8$ S, effectively reducing the ionosphere's electric fields to nearly zero. Under these conditions, the ionosphere's convection electric field in principle has no impact on magnetosphere convection. However, as shown in Fig. S3a, b, the magnetosphere's convection pattern remains largely the same, with a reduced strength of approximately 70–80% (Fig. S3d). Thus, a substantial portion (~70–80%) of the magnetosphere convection electric field persists, regardless of the contribution from ionosphere electrostatic field mapping. This suggests that the primary source of magnetosphere convection might be electromagnetic and originates outside the ionosphere in the global simulation.

### Context of the study results
Based on ionosphere convection, ECPC models of the Dungey cycle predict that magnetosphere convection can be solely driven by dayside reconnection[20]. Magnetosphere convection directly driven by dayside reconnection is also proposed as a key component in the strongly-driven substorm model[14]. In this study, we present direct evidence for dayside-driven magnetosphere convection through global simulations and observations.

In our analysis of a strong solar wind case study, we demonstrate that an enhanced dayside-driven magnetosphere convection pattern emerges within 10 minutes following a southward turning of the IMF. Our global simulations reveal a 10–20 minute progression of the enhanced magnetosphere convection and Region 1/2 FAC from the dayside towards the nightside. This result is achieved through self-consistent computation of the temporal evolution of magnetosphere convection, ionosphere convection electric fields, Region 1/2 FAC, and dayside reconnection. Observational evidence within this short timescale also reveals enhancements in both magnetosphere convection and the ionosphere's two-cell convection (DP-2). The 10-20 minutes timescale of progression is consistent with estimates from previous models[14,20].

Our findings support the paradigm that dayside reconnection and nightside reconnection act as two independent drivers for magnetosphere convection. The results highlight the importance of Region 1/2 FAC in the global-scale coupling of solar wind, magnetosphere, and ionosphere. A conjunction measurement of Region 1/2 FAC and convection would enhance the scientific exploration of the upcoming Solar-Wind-Magnetosphere-Ionosphere Link Explorer (SMILE) mission[44].

## Methods
### The PPMLR-MHD Global simulation
The global simulations are performed using the piecewise parabolic method with an extended Lagrangian remap (PPMLR)-MHD algorithm[45,46]. The MHD equations are solved in the Geocentric Solar Magnetospheric (GSM) coordinate system in the region of $-100R_E < X < 25R_E$, $-50R_E < Y, Z < 50R_E$. A uniform mesh with a spacing of $0.2\ R_E$ is used in $-15R_E < X, Y, Z < 15R_E$, with a gradually-increasing grid size outside the cube along $X, Y, Z$ axis. The inner boundary of the magnetosphere is at radial distance 3 $R_E$, confining an ionosphere domain of invariant latitude > 54°. The ionosphere conductivity is from the empirical mode in[47]. For comparison, we also conduct a run with an infinite ($10^8 S$) ionosphere conductivity so that the ionosphere electric fields and potential are nearly zero. The magnetosphere-ionosphere coupling follows the scheme in[25,35,46], involving a mapping of FAC from the magnetosphere inner boundary to the ionosphere and convection electric fields from the ionosphere to the magnetosphere inner boundary. The time step for the simulation run is set to 1 minute. The interplanetary parameters input to the MHD simulations is the 1-minute definitive data from OMNI.

### Data from Interplanetary measurements
Interplanetary observation data comes from the 1-minute definitive data from OMNI. In this event, the OMNI data is based on a time-shift of WIND observations to the Earth bow shock (-10$R_E$). We make a minor correction on the arrival time of the IMF southward turning from 12:41:30UT (OMNI) to 12:43:00UT (Fig. 2a1), based on a visual inspection of magnetic field data from Cluster 4 near Earth's bow shock (supplementary Fig. S4).

### Data from magnetosphere measurements
Observation data from the magnetosphere include bulk flow velocity, and ion energy spectrum from Hot Plasma Composition Analyzer (HPCA)[48] on MMS spacecraft[49].

### Data from ionosphere measurements
Observation data from ionosphere include 1-minute Polar Cap (PC-N) index[50,51], the AU/AL index[52], cross-polar-cap potential and the ionospheric convection maps from Super Dual Auroral Radar Network (SuperDARN)[28,53,54]. The PC-N index estimated the intensity of anti-sunward plasma convection in the high-latitude polar caps by measuring the magnetic perturbations of the Hall current. Similarly, the AU/AL index measured the strength of the Hall current associated with the return convection in the low-latitude aurora zone. Additionally, the AL index also measured intensified westward electric current in the aurora zone associated with substorms. The production of ionospheric convection maps is described in[27,55]. Line-of-Sight (LOS) velocities from available radars are processed, filtered, and binned onto a uniform spatial grid. A statistical convection model[56] is employed to derive the ionospheric convection map, following the procedure detailed in[27]. The total cross-polar cap potential is determined from the ionospheric convection maps.

### Reporting summary
Further information on research design is available in the Nature Portfolio Reporting Summary linked to this article.

## Data availability

The MHD simulation data (including the input and output) generated in this study have been deposited at https://zenodo.org/records/10429719. OMNI and MMS data are available at NASA's Coordinated Data Analysis Web (CDAWeb, http://cdaweb.gsfc.nasa.gov/). Super-DARN data can be accessed at https://doi.org/10.20383/102.0446. The data and the procedure of SuperDARN convection maps for this event study is available at https://zenodo.org/records/10374021. All data supporting the findings of this study are available from the authors on request.

## Code availability

The radar software toolkit (RST) used to produce SuperDARN convection maps is available at https://doi.org/10.5281/zenodo.7467337, and reference therein. The PPMLR-MHD simulation code is available from the corresponding author upon request. SPEDAS codes used for analyzing MMS data are freely available at http://spedas.org/blog/.

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

## Acknowledgements

Lei Dai is grateful for the insights and encouraging comments from Professor Charles Kennel. We acknowledge the use of SuperDARN data. SuperDARN is a network of radars funded by national scientific funding agencies of Australia, Canada, China, France, Italy, Japan, Norway, South Africa, the United Kingdom, and the United States of America. The work at NSSC was supported by NNSFC grants (42188101, 42174207), the Specialized Research Fund for State Key Laboratories of China, and the Strategic Pionner Program on Space Science II, Chinese Academy of Sciences, grants XDA15350201, XDA15052500. W. G. would like to thank the support given by the China-Brazil Joint Laboratory for Space Weather (CBJLSW) in Brazil.

## Author contributions

L.D. contributed to the conceptualization of the study, data analysis, data interpretation, and the manuscript's writing. M.Z. and B.T. conducted the global MHD simulations. M.Z. and A.S. contributed to analyzing simulations data. Y.R. and J.Z. contributed to analyzing the observation data. W.G., D.S., A.S., and P.E. contributed to data interpretation. C.W. and G.B.-R. contributed to the revision of the manuscript. All authors participated in the manuscript reviewing and editing.

## Competing interests

The authors declare no competing interests.
