## [Peer Review File · Nature Communications]

Global-scale magnetosphere convection driven by dayside magnetic reconnectionReviewers' comments:

Reviewer #1 (Remarks to the Author):

Paper Review for "Driving Magnetosphere Convection through the ionosphere" by Dai et al

In this paper, the authors present evidence of magnetospheric/ionospheric convection driven purely by dayside reconnection. They present evidence from MHD simulations and also some observational evidence. Although the paper reads well, there is some work to be done to make it ready for publication. However, once the suggested changes have been made I feel it will be an excellent addition to the literature.

Major comments:

1. From reading the paper, the model for how dayside reconnection drives global convection is not entirely clear to me. Perhaps the addition of a detailed description of the model in the conclusion will make this clear.
2. The paper is missing a "data" section. Please include this section after the introduction and include: your description of the MHD from the appendix, and a description of what all the observational data are with relevant data citations. In particular ensure you include processing options for the SuperDARN convection maps as this will ensure the repeatability of your work.
3. I feel the order of things could be improved – the observational data should come before the interpretation and discussion, to justify Section 2.2 "dayside magnetic reconnection is the only driver"
4. Finally, your description of the figures can be quite light. When introducing the figure, please explain what it shows in detail, and then also be more detailed with your description of what signatures you are picking out from the figure.
5. The introduction is not quite thorough enough. Please include a description of the canonical morphology of FACs/ionospheric convection/magnetospheric convection to improve interpretability of your results.

Detailed/minor comments:

- Introduction, paragraph 2, "over a timescale of approximately one hour", please include a citation for this
- Introduction, paragraph 3, "through the field-aligned current (FAC) and Alfvén waves". Please explain this in more detail, how do the FACs and Alfvén waves control/drive the convection pattern? My understanding is that it is not clear as to whether FACs drive convection or convection drives FACs. Please elaborate.
- Introduction, paragraph 3, "approximately 10 minutes" is this to suggest that the entire cycle of convection, tracing a particular field line, occurs in 10 minutes? – this is remarkably quick given how slow the ionosphere is to respond to large scale changes.
- Introduction, paragraph 4, "the ionosphere's electric fields (or FAC) determined by dayside reconnection" are they? What would a purely dayside reconnection driven FAC look like? Please can you describe this in more detail, linking to citations.
- Introduction, I feel that the introduction spends a while focusing on the new model that the authors present, and there isn't quite enough literature review/citations. Please could the authors expand the literature review, I suggest some more citations on the morphology of the FACs generally, and under purely dayside driving. You should cite canonical FAC references Iijima and Potemra (below); also should probably cite the diagrams of Provan et al 2004 (doi 10.5194/angeo-22-3607-2004)
 - o Iijima, T. and Potemra, T. A. Field-aligned currents in the dayside cusp observed by Triad. *Journal of Geophysical Research*, 81(34), 1976a. doi: 10.1029/JA081i034p05971.
 - o Iijima, T. and Potemra, T. A. The amplitude distribution of field-aligned currents at northern high latitudes observed by Triad. *Journal of Geophysical Research*, 81(13), 1976b. doi: 10.1029/JA081i013p02165.
 - o Iijima, T. and Potemra, T. A. Large-scale characteristics of field-aligned currents associated with substorms. *Journal of Geophysical Research: Space Physics*, 83(A2), 1978. doi: 10.1029/JA083iA02p00599.
- Fig 2(a3) v_x is clearly the dominant component of velocity in this interval. Because of the difference in magnitude between v_x and other components, you can't really see the variation on

vx. I suggest presenting just vx with an appropriate yscale so that you can see the variation (or indeed, lack of).

- Fig 2(b-e) the convention for ionospheric plots is to have noon at the top, midnight at the bottom, dawn on the right and dusk on the left. Please consider rotating the plot to this convention – this will increase interpretability of your work.
- Section 2.1, paragraph 1 / Fig 2(b-e) the region 2 FACs are very hard to identify, please could you saturate the colour bar. I feel that this description of FACs comes out of the blue because you haven't described their general morphology in the intro – see my previous point. Also, the FACs in your figure are in the opposite sense to the literature, and indeed your description. See e.g. Milan et al 2017 doi: 10.1007/s11214-017-0333-0.
- Section 2.1, paragraph 1, "the extension to the nightside is clear" please describe the morphology of this extension – it will not be clear to everyone!!
- Fig 2 b2-e2, I can only see two lines mapping invariant latitude, but the caption says there are three.
- Section 2.1, paragraph 1. I feel that the description of Fig 2 is rather light. You should introduce the figures i.e. "Fig b1 shows a magnetic latitude-magnetic local time map of..." and similar for b2.
- Section 2.1, paragraph 2, again please explain in detail what an ewogram shows.
- Section 2.1, paragraph 2, "in general, magnetosphere convection..." citation for this needed.
- Section 2.1, final paragraph, "private communications" note to editor – are private communications allowed?
- Section 2.2 "dayside magnetic reconnection is the only driver" you have not presented evidence for this. Please make sure you rule out nightside reconnection. Perhaps you could present index AL (which you can also retrieve from OMNI) and show whether there are signatures of substorms in that 30 minutes? NOTE: I see later in the paper you have done this, this needs to come before the 2.2 section.
- Fig 3, caption description of a1/a2/b1/b2 doesn't match the figure
- Section 2.2, paragraph 2, "private communications" – note to editor- are private communications allowed?
- Section 2.3, paragraph 1. Please at this point explain the figure, e.g. "Fig. 4(a) shows the temporal evolution of the northern hemisphere polar cap index..." etc. in particular highlighting the meaning of the red shaded box and how it relates to your other figures.
- Section 2.3, paragraph 2. Please include a citation for each dataset: For PC-N (Stauning (2013); Troshichev et al. (1979); Troshichev and Andrezen (1985)), for AU/L (Davis & Sugiura, 1966)
- Section 2.3, paragraph 2. "AU index is associated with the strength of low-latitude two-cell convection in the aurora zone" citation for this please.
- Section 2.3, paragraph 2. "This ionosphere 2D convection..." please describe this figure in detail e.g. "each panel shows a magnetic latitude-magnetic local time map of ionospheric convection..." describe what the flagpoles/contours show.
- Section 2.3, paragraph 2. "This ionosphere 2D convection..." Additionally, please describe what signatures you are seeing with respect to your timeline. You have not described the results from these panels
- Section 2.3, paragraph 2. "additional ionosphere convection cell" please note where this is and in which panel.
- Section 2.3, paragraph 3. What about the dip in AL that occurs half way through the red box? Although small, this could be evidence of a pseudo-breakup/mini-substorm. Please discuss this.
- Conclusion, please also discuss what you have learnt from fig. 4
- Conclusion, please describe, in detail, what is your model for how dayside reconnection drives
- Data availability: please include where you accessed SuperDARN convection maps and MMS data.

Reviewer #2 (Remarks to the Author):

The manuscript describes an alternative explanation for the magnetosphere convection during southward IMF. The main claim is that the convection is driven by the dayside reconnection and the ionosphere in the first 30 minutes. The authors provide numerical modeling and observational support for their theory.

Overall, this is a well written work with clear explanations, at least for those with significant knowledge of the field.

I have two significant comments/requests. The first one is related to the modeling. I suggest to conduct an experiment where the MHD model is run without the ionosphere model (using some simple boundary conditions at 3Re) and determine the initial flow patterns and how they deviate (or not) from those obtained with the coupled model. In the experience of this reviewer, the feedback of the ionosphere model to the magnetosphere model is rather weak, and it may well be that the flow patterns are generated purely by the MHD model. One could also conduct experiments with some simplified ionospheric conductance model, for example constant conductance. But the most important question is if the ionospheric convection is the driver or simply a consequence of the magnetospheric flows. This question can be settled with the proposed simulation.

Going to the observations, it was not clear how the ionospheric convection maps are obtained. Are those from observations or from an empirical model, or from simulations? This is important, because the convection maps are shown to provide observational support for the theory. It should be made clear to the reader which part of Figure 4f-k is actual observation.

Reviewer #3 (Remarks to the Author):

Review of "Driving Magnetosphere Convection through the ionosphere" by Dai et al.

This paper claims to have uncovered a new driving process whereby plasma convection in the closed part of the magnetosphere can be driven by dayside reconnection alone. Unfortunately, this is not a new concept, and has in fact been a widely accepted component of magnetospheric driving for many decades. The Expanding Contracting Polar Cap (ECPC) model (Cowley and Lockwood, 1992) explicitly requires the convection of closed flux from the nightside in response to dayside reconnection (even with zero nightside reconnection). See, e.g. Figure 6a of that paper. Numerous studies have since further discussed the ECPC, with a summary provided in a recent review by Milan and Grocott (2021) in which they explicitly state that "there is a general sunwards movement of flux in the closed magnetosphere" as a consequence of dayside reconnection with zero nightside reconnection. In other words, the idea that convection of closed magnetospheric flux may be excited by dayside reconnection is an established concept.

Without the novelty of a new discovery, the paper offers little beyond some rather limited evidence in support of an already well-established concept. As such, I do not consider the paper to be suitable for publication in Nature Communications. If the study was developed such that it revealed some novel details that might augment our understanding, then it might become suitable for publication. To that end, I offer some further comments on the content of the paper.

The authors suggest that it is magnetosphere-ionosphere coupling that is crucial to the ability of dayside reconnection to drive convection in the magnetosphere. To an extent this suggestion is true, in that the ionosphere allows magnetospheric currents to close, thereby allowing magnetospheric convection to occur (Ridley et al, 2004). But this is not unique to the case of unbalanced dayside reconnection.

The simulation results of Figure 2 certainly seem consistent with the ECPC model, but they do not show anything unexpected. See, e.g. Figure 2i of Milan (2013), which shows theoretical model results for an interval of dayside reconnection, depicting largely the same thing.

I find the results presented in Figure 3 a little hard to interpret. The positive/negative slopes apparent after the dashed vertical line seem to also be present before the line. But in any case, that (as the authors state) these data "demonstrate the clear relation between the ionosphere's FAC and the magnetosphere's convection" is wholly expected based on current understanding.

The authors cite a private communication with Charles Kennel when suggesting that the relative strength of dayside and nightside driven convection is likely determined by the value of cross-polar-cap electric potential. There are many publications that could be cited to demonstrate the relationship between dayside and nightside driven convection and the value of cross-polar-cap electric potential. In fact, it is the strength of the cross-polar-cap electric potential that is determined by the relative strength of the dayside and nightside driven convection (e.g. Milan and Grocott, 2021 and references therein).

The authors present SuperDARN convection maps as evidence for the ionospheric counterpart of the magnetospheric flow, depicting enhanced flow being driven first on the dayside immediately following the southward IMF turning, reaching the nightside some minutes later. Active nightside convection on closed field lines during (expected) dayside reconnection driving is not a new observation, e.g. Figure 6a of Grocott et al. (2002) shows modestly active flows in the nightside closed field line region during the growth phase of a substorm (i.e. prior to onset of nightside reconnection). It's important to note though, that in this case (and in the case of the paper under review) it is not possible to rule out some low-level nightside reconnection being active during this time – intervals of true “zero nightside reconnection” are rare and hard to identify.

Cowley, S. W. H. & Lockwood, M. Excitation and decay of solar wind-driven flows in the magnetosphere-ionosphere system. *Annales Geophysicae* 10, 103–115 (1992).

Grocott, A., Cowley, S. W. H., Sigwarth, J. B., Watermann, J. F. & Yeoman, T. K. Excitation of twin-vortex flow in the nightside high-latitude ionosphere during an isolated substorm, *Annales Geophysicae* 20: 1577–1601, (2002).

Ridley, A. J., Gombosi, T. I. & DeZeeuw, D. L. Ionospheric control of the magnetosphere: conductance, *Annales Geophysicae* 22: 567–584 (2004).

Milan, S. E. Modeling Birkeland currents in the expanding/contracting polar cap paradigm, *J. Geophys. Res. Space Physics*, 118, 5532–5542, doi:10.1002/jgra.50393 (2013).

Milan, S. E. & Grocott, A. High latitude ionospheric convection, in *Space Physics and Aeronomy Collection Volume 3: Ionosphere Dynamics and Applications*, Geophysical Monograph 260, eds. C. Huang 10 and G. Lu, American Geophysical Union, Wiley and Sons, 21-47, <https://doi.org/10.1002/9781119815617.ch2> (2021).

We greatly appreciated all reviewers for comments that help significantly improve the manuscript. We have substantially revised the manuscript in light of your comments. The following are detailed one-to-one responses.

Reviewers' comments:

Reviewer #1 (Remarks to the Author):

Paper Review for "Driving Magnetosphere Convection through the ionosphere" by Dai et al

In this paper, the authors present evidence of magnetospheric/ionospheric convection driven purely by dayside reconnection. They present evidence from MHD simulations and also some observational evidence. Although the paper reads well, there is some work to be done to make it ready for publication. However, once the suggested changes have been made I feel it will be an excellent addition to the literature.

Major comments:

1. From reading the paper, the model for how dayside reconnection drives global convection is not entirely clear to me. Perhaps the addition of a detailed description of the model in the conclusion will make this clear.

Thank you for your feedback. In the previous version, we initially proposed a driving mechanism that exclusively involves the ionosphere. In this model, the ionosphere's convection electric field was thought to directly penetrate to low latitudes and subsequently map back to the magnetosphere, driving magnetospheric convection. However, this model introduced several uncertainties that warranted further investigation. Consequently, we discussed this mechanism solely within the discussion section.

In our revised version, we conduct additional simulation runs that make us re-consider the driving mechanism. In these simulations, we observed that global magnetospheric convection still occurs even when the ionosphere's electric field was set to zero (with an infinite ionospheric conductivity). This outcome compelled us to reevaluate our previous interpretation. We now propose that the driving mechanism is mainly a result of magnetospheric processes, with the Region 1 Field-Aligned Current (FAC) playing a significant role by reducing the magnetic pressure on the dayside magnetopause.

There remain uncertainties within this updated model that necessitate further investigation. Therefore, we have maintained the detailed description of the driving mechanism in the discussion section 4. In the conclusion, we simply acknowledge the close association between the driving force and the Region 1 FAC. Our focus in this manuscript is primarily on presenting direct evidence supporting the concept that dayside reconnection drives global convection. We hope this refined approach is

acceptable.

2. The paper is missing a “data” section. Please include this section after the introduction and include: your description of the MHD from the appendix, and a description of what all the observational data are with relevant data citations. In particular ensure you include processing options for the SuperDARN convection maps as this will ensure the repeatability of your work.

Thank you for your comment. We have addressed this by adding a data section 2 following the introduction. In this section, we provide a comprehensive description of the MHD data and the observational data sources with citations. Additionally, we outline the processing method for the SuperDARN convection maps. Modifications are in Line 185-255.

3. I feel the order of things could be improved – the observational data should come before the interpretation and discussion, to justify Section 2.2 “dayside magnetic reconnection is the only driver”

Thank you for this comment. We have modified the order of the section accordingly. Observational data is now moved to before the interpretation and discussion.

4. Finally, your description of the figures can be quite light. When introducing the figure, please explain what it shows in detail, and then also be more detailed with your description of what signatures you are picking out from the figure.

Thank you for your input. We've addressed this issue by providing more detailed descriptions for each figure in the revised manuscript. These descriptions now include specific references to individual panels, enhancing the clarity of the figures. These improved descriptions are throughout the revised text.

5. The introduction is not quite thorough enough. Please include a description of the canonical morphology of FACs/ionospheric convection/magnetospheric convection to improve interpretability of your results.

Responses: Thank you for this comment. We have added description of the Region 1/2 FAC and the two-cell convection. Lines 125-133.

Detailed/minor comments:

- Introduction, paragraph 2, “over a timescale of approximately one hour”, please include a citation for this

We add a reference for this statement. Line75.

- Introduction, paragraph 3, “through the field-aligned current (FAC) and Alfvén waves”. Please explain this in more detail, how do the FACs and Alfvén waves control/drive the convection pattern? My understanding is that it is not clear as to whether FACs drive convection or convection drives FACs. Please elaborate.

Thank you for your feedback. We've updated the text in response to this comment. In

the revised version, our interpretation has changed regarding the relationship between low-latitude ionosphere convection and magnetosphere convection. We've removed this statement in paragraph 3 in the revision.

- Introduction, paragraph 3, “approximately 10 minutes” is this to suggest that the entire cycle of convection, tracing a particular field line, occurs in 10 minutes? – this is remarkably quick given how slow the ionosphere is to respond to large scale changes. Thank you for your comment. We agree that clarifying the timescale of the convection cycle is important for a better understanding.

The entire convection cycle for a particular field line, as estimated by previous studies, typically spans 3-5 hours. The 10-minute timescale mentioned in our study specifically refers to a rapid phase propagation of an 'enhanced convection' state. Within this brief 10-minute window, the plasma associated with a particular field line only moves a relatively small distance in the convection cycle. This can explain the remarkably quick (10-minute) response of ionospheric convection to changes in the southward Interplanetary Magnetic Field (IMF). We have added this clarification in the introduction (Line 118-121).

- Introduction, paragraph 4, “the ionosphere’s electric fields (or FAC) determined by dayside reconnection” are they? What would a purely dayside reconnection driven FAC look like? Please can you describe this in more detail, linking to citations. Thank you for your comment. We have revised the text to improve clarity.

In Section 3 of the revised version, we provide a straightforward description of the Region 1 Field-Aligned Current (FAC) enhancement driven by dayside reconnection. The process involves reconnected field lines from the dayside being dragged by the anti-sunward convection of the solar wind. These dragged field lines wrap around the magnetopause, forming a magnetic shear with the closed-field lines of the magnetosphere. Near the dawn and dusk sector, the primary contribution to this magnetic shear is from the $\Delta B_x/\Delta Y$ component. This magnetic shear introduces Region 1 FAC. Modifications are in Line 425-434.

I learned about this elegant picture from David Sibeck during the recent ICS-15 International Substorm conference. We have now added Sibeck as a co-author for this valuable contribution. However, it is difficult to find an explicit citation or reference for this picture (communications with David Sibeck). Maybe this is his original idea.

- Introduction, I feel that the introduction spends a while focusing on the new model that the authors present, and there isn't quite enough literature review/citations. Please could the authors expand the literature review, I suggest some more citations on the morphology of the FACs generally, and under purely dayside driving. You should cite canonical FAC references Iijima and Potemra (below); also should probably cite the diagrams of Provan et al 2004 (doi 10.5194/angeo-22-3607-2004)

o Iijima, T. and Potemra, T. A. Field-aligned currents in the dayside cusp observed by Triad. *Journal of Geophysical Research*, 81(34), 1976a. doi: 10.1029/JA081i034p05971.

o Iijima, T. and Potemra, T. A. The amplitude distribution of field-aligned currents at northern high latitudes observed by Triad. *Journal of Geophysical Research*, 81(13), 1976b. doi: 10.1029/JA081i013p02165.

o Iijima, T. and Potemra, T. A. Large-scale characteristics of field-aligned currents associated with substorms. *Journal of Geophysical Research: Space Physics*, 83(A2), 1978. doi: 10.1029/JA083iA02p00599.

Responses: We add the description of Region 1/2 FAC, these relevant reference, and also the citation on diagrams of Provan et al 2004.

- Fig 2(a3) v_x is clearly the dominant component of velocity in this interval. Because of the difference in magnitude between v_x and other components, you can't really see the variation on v_x . I suggest presenting just v_x with an appropriate yscale so that you can see the variation (or indeed, lack of).

We agree that v_x is indeed the dominant component. We've incorporated this point into Line XXX of the revised text. This point is also illustrated in Supplementary Figure Fig.1S. Both v_x and the azimuthal convection velocity v_a exhibit similar evolutionary patterns, except for their signs.

Furthermore, it's also important to consider v_y to understand the flow pattern on the dayside, particularly in the MLT range from 9 to 15. v_y indicates that the existence of convection electric field E_x , in addition to E_y . The azimuthal convection velocity v_a effectively represents the combined effects of both v_x and v_y . In the revision, we have included v_a in Figure 2 and V_x/V_y and Fig.1S.

Modifications are in line 285-288 and Fig.2S. We hope these modifications address the comment effectively.

- Fig 2(b-e) the convention for ionospheric plots is to have noon at the top, midnight at the bottom, dawn on the right and dusk on the left. Please consider rotating the plot to this convention – this will increase interpretability of your work.

We have rotated Fig.2(b-e) accordingly.

- Section 2.1, paragraph 1 / Fig 2(b-e) the region 2 FACs are very hard to identify, please could you saturate the colour bar. I feel that this description of FACs comes out of the blue because you haven't described their general morphology in the intro – see my previous point. Also, the FACs in your figure are in the opposite sense to the literature, and indeed your description. See e.g. Milan et al 2017 doi: 10.1007/s11214-017-0333-0.

We have implemented the suggestion to lower the upper limit of the color bar. As a result, Region 2 Field-Aligned Currents (FACs) are now more discernible in Figure 2. Additionally, we have included a reference to Region 1/2 FAC in the introduction. Region 1 FAC flows into the ionosphere at dawn and out of the ionosphere at dusk, while Region 2 FAC exhibits an opposite direction to that of Region 1 FAC. We have

verified that the direction of the FACs are consistent with that in the existing literature.

- Section 2.1, paragraph 1, “the extension to the nightside is clear” please describe the morphology of this extension – it will not be clear to everyone!!

We agree that the signal of Region 2 FAC was not very clear in the previous version. In response to this feedback, we have refined the color scale to enhance the visibility of these signals. In addition, we also have adjusted the description as follows: "The progression of Region 2 FAC and the dawn part of Region 1 FAC towards the nightside is clear (Fig.2.c1-e1). Modifications are in Line 277-280.

- Fig 2 b2-e2, I can only see two lines mapping invariant latitude, but the caption says there are three.

The map of the 75-degree invariant latitude sometimes extends beyond the closed region of the magnetosphere and may not always be displayed clearly. The third line may be easier to be seen in panel b2.

- Section 2.1, paragraph 1. I feel that the description of Fig 2 is rather light. You should introduce the figures i.e. “Fig b1 shows a magnetic latitude-magnetic local time map of...” and similar for b2.

We have modified the description accordingly. Modifications are in Line 270-273.

- Section 2.1, paragraph 2, again please explain in detail what an ewogram shows. We have added more description of ewogram. The ewogram displays a MLT-time map of a quantity averaged in a certain range of invariant latitudes. Modifications are in Line 292-295.

- Section 2.1, paragraph 2, “in general, magnetosphere convection...” citation for this needed.

The statement in the previous version seems to be confusing. It was intended to describe the results in the simulations. We have revised the statement : "In this interval, magnetosphere convection associated with Region-2 FAC is weaker than that associated with Region-1 FAC (Fig3.b1-b2)." Modifications are in line 307-309.

- Section 2.1, final paragraph, “private communications” note to editor – are private communications allowed?

We have leaved the issue to be decided later by the editor.

- Section 2.2 “dayside magnetic reconnection is the only driver” you have not presented evidence for this. Please make sure you rule out nightside reconnection. Perhaps you could present index AL (which you can also retrieve from OMNI) and show whether there are signatures of substorms in that 30 minutes? NOTE: I see later in the paper you have done this, this needs to come before the 2.2 section.

We agree with this view. It is difficult to completely rule out nightside reconnection in observations because we do not have global-scale spacecraft coverage.

In this study, we only rule out nightside reconnection in global simulations. In simulations, we can use v_x and v_a (Fig.2 and Fig.1S) to identify that outflow from nightside reconnection is absent in the interval $T < 10$ min. Additionally, Fig.3 also shows a dayside-driven convection that cannot be explained by nightside reconnection. Convection driven by nightside reconnection needs to be initiated in the midnight and progress toward the dayside, which is not the case in Fig.3. We have added clarification in Line 333-338 in section 3.1.

- Fig 3, caption description of a1/a2/b1/b2 doesn't match the figure
We have made correction accordingly.

- Section 2.2, paragraph 2, "private communications" – note to editor- are private communications allowed?
We have removed it since we have changed the interpretation.

- Section 2.3, paragraph 1. Please at this point explain the figure, e.g. "Fig. 4(a) shows the temporal evolution of the northern hemisphere polar cap index..." etc. in particular highlighting the meaning of the red shaded box and how it relates to your other figures.
We have made the corresponding modification and added more explanation.

In Fig.4, we present magnetosphere-ionosphere observations related to the dayside-driven convection event on March 11. Panels a-e display the temporal evolution of the northern hemisphere polar cap index, AU/AL index, the cross-polar-cap potential, magnetosphere convection, and the ion energy spectrum from MMS. The red shaded box marks the first interval ($T=1-9$ min) of the enhanced convection, as analyzed in simulations in Fig. 2 and Fig. 3. The blue shaded box marks the interval of the second convection enhancement. Panels f-k shows the ionosphere convection maps in these intervals. We focus on two aspects in observations." Modifications are in line 350-361.

- Section 2.3, paragraph 2. Please include a citation for each dataset: For PC-N (Stauning (2013); Troshichev et al. (1979); Troshichev and Andrezen (1985)), for AU/L (Davis & Sugiura, 1966)
We have now added the citation for the data set in the Data section. Modifications are in line 231-235.

- Section 2.3, paragraph 2. "AU index is associated with the strength of low-latitude two-cell convection in the aurora zone" citation for this please.
The reference can be found in paragraph 3.3 of Kepko et al., 2015SSR. The AU index is a rough measure of the DP-2 current, representing two-cell convection in the aurora zone. We've added this reference. Modifications are in line 372-374.

- Section 2.3, paragraph 2. "This ionosphere 2D convection..." please describe this figure in detail e.g. "each panel shows a magnetic latitude-magnetic local time map of

ionospheric convection..." describe what the flagpoles/contours show.

We add more description of the convection map. Panel f-k shows the invariant latitude-MLT map of ionospheric convection (colored flagpoles) and electrostatic potential (contours). Modifications are in Line 358-361.

- Section 2.3, paragraph 2. "This ionosphere 2D convection..." Additionally, please describe what signatures you are seeing with respect to your timeline. You have not described the results from these panels

We've added more detailed descriptions of the results in the ionosphere 2D convection map.

The first four snapshots of the ionosphere convection map (Fig. 4f-i) also depict a rise in convection (approximately 500-1000 m/s) at latitudes greater than 65 degrees. Notably, the most significant enhancement (1000 km/s) progresses from MLT=8-9 towards MLT=7 between 12:48 UT and 12:52 UT in Fig. 4h-i. The second enhancement of convection occurs during T=19-23 minutes (marked by the blue box), coinciding with a substorm expansion denoted by a sharp decrease in the AL index. Correspondingly, the ionosphere convection map (Fig. 4j-k) reveals an enhancement (1000 km/s) in convection velocity at latitudes greater than 60 degrees. Modifications are in Line 377-389.

- Section 2.3, paragraph 2. "additional ionosphere convection cell" please note where this is and in which panel.

We add the information in the supplementary Fig.S2.

- Section 2.3, paragraph 3. What about the dip in AL that occurs half way through the red box? Although small, this could be evidence of a pseudo-breakup/mini-substorm. Please discuss this.

We have added further discussion on this point. The increase in AU and AL is similar during T=1-9 minutes, suggesting an enhanced two-cell convection (DP-2) without the presence of the substorm current system (DP-1). From this, we can conclude that the substorm current system (SWC/DP-1) is absent during this interval. Otherwise, SWC/DP-1 (westward electrojet) would induce an AL much larger than AU (eastward electrojet associated with two-cell convection). Modifications are in Line 375-377.

- Conclusion, please also discuss what you have learnt from fig. 4

We have incorporated more details about the observational results into the conclusion. "Observational evidence within this short timescale (10-20 minutes) also reveals enhancements in both magnetosphere convection and the ionosphere's two-cell convection." Modifications are in line 523-526.

- Conclusion, please describe, in detail, what is your model for how dayside reconnection drives

In the conclusion, we've added more details of how dayside reconnection drives magnetosphere convection. The driving of magnetosphere convection is closely linked to the enhancement of Region 1 FAC resulting from dayside reconnection. More details of the models is described in the interpretation section. Line422-426

- Data availability: please include where you accessed SuperDARN convection maps and MMS data.

Responses: We include more information on the data availability. Line 554-563 MMS data are available <http://cdaweb.gsfc.nasa.gov/>. SuperDARN data can be accessed at <http://vt.superdarn.org/tiki-index.php?page=Data+Access>. SuperDARN convection maps are produced from radar software toolkit (RST 4.3) which is available at Zenodo <https://doi.org/10.5281/zenodo.3401622>

Reviewer #2 (Remarks to the Author):

The manuscript describes an alternative explanation for the magnetosphere convection during southward IMF. The main claim is that the convection is driven by the dayside reconnection and the ionosphere in the first 30 minutes. The authors provide numerical modeling and observational support for their theory.

Overall, this is a well written work with clear explanations, at least for those with significant knowledge of the field.

I have two significant comments/requests. The first one is related to the modeling. I suggest to conduct an experiment where the MHD model is run without the ionosphere model (using some simple boundary conditions at 3Re) and determine the initial flow patterns and how they deviate (or not) from those obtained with the coupled model. In the experience of this reviewer, the feedback of the ionosphere model to the magnetosphere model is rather weak, and it may well be that the flow patterns are generated purely by the MHD model. One could also conduct experiments with some simplified ionospheric conductance model, for example constant conductance. But the most important question is if the ionospheric convection is the driver or simply a consequence of the magnetospheric flows. This question can be settled with the proposed simulation.

We greatly appreciate this comment. In response to your suggestion, we conducted a test simulation for this same event with a very high ionosphere conductivity (10^8 S), effectively reducing ionosphere electric fields and convection to zero. In this scenario, the ionosphere's convection electric field had no impact on magnetospheric convection. The results of this test run are summarized in supplementary Fig. S3. As seen in Fig. S3a-b, the overall pattern of magnetospheric convection remained largely consistent, with a reduced strength of approximately 70-80% (Fig. S3d). This finding indicates that a significant proportion (70-80%) of the

magnetospheric convection electric field persists independently of ionosphere electrostatic field mapping. Consequently, it implies that the primary source of magnetospheric convection is likely electromagnetic and originates beyond the ionosphere in our global simulation.

These results have led to a substantial reinterpretation of our findings. This reinterpretation is reflected in the revised title, abstract, and conclusion. While the ionosphere does indeed influence magnetospheric convection through its conductivity, the driving process is primarily within the magnetosphere. In section 4, we provide a simplified model of how dayside reconnection propels convection through the Region 1 Field-Aligned Current (FAC). Modifications include the title, abstract, conclusion, and line 484-504.

Going to the observations, it was not clear how the ionospheric convection maps are obtained. Are those from observations or from an empirical model, or from simulations? This is important, because the convection maps are shown to provide observational support for the theory. It should be made clear to the reader which part of Figure 4f-k is actual observation.

Thank you for your comment. The ionospheric convection maps are mainly observations, complemented by fitting a statistical convection model. We have included details about the method for generating these ionospheric convection maps in section 2.2. Modification are in line 243-255.

Reviewer #3 (Remarks to the Author):

Review of "Driving Magnetosphere Convection through the ionosphere" by Dai et al.

This paper claims to have uncovered a new driving process whereby plasma convection in the closed part of the magnetosphere can be driven by dayside reconnection alone. Unfortunately, this is not a new concept, and has in fact been a widely accepted component of magnetospheric driving for many decades. The Expanding Contracting Polar Cap (ECPC) model (Cowley and Lockwood, 1992) explicitly requires the convection of closed flux from the nightside in response to dayside reconnection (even with zero nightside reconnection). See, e.g. Figure 6a of that paper. Numerous studies have since further discussed the ECPC, with a summary provided in a recent review by Milan and Grocott (2021) in which they explicitly state that "there is a general sunwards movement of flux in the closed magnetosphere" as a consequence of dayside reconnection with zero nightside reconnection. In other words, the idea that convection of closed magnetospheric flux may be excited by dayside reconnection is an established concept.

Thank you for your comment. In the previous version, we did not adequately review the literature on the ECPC model and its connection to our study. This critical aspect was missing from our manuscript, and we appreciate your feedback as it has significantly improved the context of our study's results. Consequently, we have made substantial changes in the revision.

We agree that the ECPC model introduce the concept that convection of closed magnetosphere might be solely driven by dayside reconnection. However, our approach is distinct and relies on different types of data.

It's important to note that ECPC models do not directly address magnetospheric convection. Instead, these models imply magnetospheric convection through its mapping with ionospheric convection. While this mapping is straightforward in cases of steady-state convection, it becomes more complex in the interesting scenario of non-steady convection. To the best of my knowledge, ECPC models and subsequent studies have not explored actual data of magnetosphere convection [e.g., Cowley and Lockwood, 1992, Milan and Grocott (2021)].

In our approach, we utilize global simulations to provide real data on magnetospheric convection. In the global simulations, the magnetosphere-ionosphere coupling is treated in a self-consistent manner in the case of non-steady convection. By combining these simulations with observational data on magnetospheric convection, our approach provide direct evidence for dayside-driven magnetospheric convection, which supports the ECPC paradigm.

In the revised version, we have added clarifications regarding the relationship between the ECPC model and our study. These changes are reflected in the abstract, introduction, discussion, and conclusion. Line 37-41, 87-102,139-184,445-466,509-514.

Without the novelty of a new discovery, the paper offers little beyond some rather limited evidence in support of an already well-established concept. As such, I do not consider the paper to be suitable for publication in Nature Communications. If the study was developed such that it revealed some novel details that might augment our understanding, then it might become suitable for publication. To that end, I offer some further comments on the content of the paper.

We greatly appreciate this comment as it has prompted us to reconsider the novel aspects of our approach.

Firstly, our approach uses direct data to examine magnetosphere convection, as previously mentioned. Additionally, our approach differs in how we examine the temporal evolution of dayside-driven convection.

The ECPC model relies on assumed values for the length and strength (potential drop) of dayside reconnection [Siscoe1985, Milan2013]. As a result, ECPC models require predetermined parameters for dayside reconnection. This leads to the necessity of assuming the temporal evolution of dayside reconnection when examining the temporal evolution of convection within the ECPC model, which is a challenging task. In contrast, our approach using global simulations eliminates the need for making assumptions about the temporal evolution of dayside reconnection. Our approach enables a self-consistent examination of the temporal evolution of magnetospheric convection as a response to dayside reconnection. We have incorporated these discussions into the introduction section (Line 139-184).

Our global simulations reveal a 10-minute-scale progression of magnetosphere convection and FAC from the dayside to the nightside. This is a new feature of dayside-driven magnetosphere convection. To the best of my knowledge, it is difficult, if not impossible, to obtain such temporal evolutions, especially the 10-minute timescale, using the ECPC model. The ECPC model is fundamentally a steady-state model and cannot produce a 10-minute-scale progression without assuming it. We have included these discussions in the discussion section. Modifications are in Line 445-466.

The authors suggest that it is magnetosphere-ionosphere coupling that is crucial to the ability of dayside reconnection to drive convection in the magnetosphere. To an extent this suggestion is true, in that the ionosphere allows magnetospheric currents to close, thereby allowing magnetospheric convection to occur (Ridley et al, 2004). But this is not unique to the case of unbalanced dayside reconnection.

We agree with this comment. It's known that the ionosphere influences magnetosphere convection primarily through its conductivity (Ridley et al., 2004).

In the revision, we conducted a test run, which is elaborated on in our reply to reviewer 2. During this test run, we increase the ionosphere conductivity, effectively reducing the ionosphere electric field to nearly zero. The electrostatic mapping from ionosphere to the magnetosphere becomes invalid in the test run. These updated results have led us to significantly revise our interpretation. The specific modifications can be found in Line 484-504.

The simulation results of Figure 2 certainly seem consistent with the ECPC model, but they do not show anything unexpected. See, e.g. Figure 2i of Milan (2013), which shows theoretical model results for an interval of dayside reconnection, depicting largely the same thing.

While it is true that our simulation results in Figure 2 are in agreement with the ECPC model, it's important to note that our study offers additional valuable insights, as elaborated in response to the previous comment.

First, our simulations analyze data of magnetospheric convection, in contrast to Milan (2013), which primarily focuses on data of ionospheric convection. Furthermore, our simulations investigate the temporal evolution of magnetospheric convection, dayside reconnection, and Field-Aligned Currents (FACs) in a self-consistent manner. In contrast, examining the temporal evolution of convection in Milan (2013) requires prior assumptions about the reconnection process. The results of a 10-minute-scale progression of magnetospheric convection and FAC, as presented in our study, are challenging to achieve self-consistently within the ECPC model in Milan (2013).

In the revised manuscript, we have incorporated these clarifications in Line 445-466.

I find the results presented in Figure 3 a little hard to interpret. The positive/negative slopes apparent after the dashed vertical line seem to also be present before the line. But in any case, that (as the authors state) these data “demonstrate the clear relation between the ionosphere’s FAC and the magnetosphere’s convection” is wholly expected based on current understanding.

We agree it is difficult to interpret the simulation results prior to the southward turning of IMF Bz (indicated by the dashed vertical line in the figure). In our previous version, we lacked a clear explanation for these observations.

In this revised version, we have provided more discussion regarding this observation in Line 318-328. The increase in Region 1/2 Field-Aligned Currents (FAC) might be associated with viscosity-type effects under northward IMF conditions, as described in the model in Sonnerup1980JGR (doi:10.1029/JA085iA05p02017). Dayside-reconnection certainly greatly increase Region 1/2 FAC in simulations.

The authors cite a private communication with Charles Kennel when suggesting that the relative strength of dayside and nightside driven convection is likely determined by the value of cross-polar-cap electric potential. There are many publications that could be cited to demonstrate the relationship between dayside and nightside driven convection and the value of cross-polar-cap electric potential. In fact, it is the strength of the cross-polar-cap electric potential that is determined by the relative strength of the dayside and nightside driven convection (e.g. Milan and Grocott, 2021 and references therein).

We agree with this comment. Reference (Milan and Grocott, 2021, Lockwood,1991) are added in the revision.

The authors present SuperDARN convection maps as evidence for the ionospheric counterpart of the magnetospheric flow, depicting enhanced flow being driven first on the dayside immediately following the southward IMF turning, reaching the nightside some minutes later. Active nightside convection on closed field lines during (expected)

dayside reconnection driving is not a new observation, e.g. Figure 6a of Grocott et al. (2002) shows modestly active flows in the nightside closed field line region during the growth phase of a substorm (i.e. prior to onset of nightside reconnection). It's important to note though, that in this case (and in the case of the paper under review) it is not possible to rule out some low-level nightside reconnection being active during this time – intervals of true “zero nightside reconnection” are rare and hard to identify.

We appreciate this comment. It's indeed challenging to entirely rule out the possibility of low-level nightside reconnection in our observations, primarily due to the limited coverage of in-situ data. However, based on our observational data, the dominant convection pattern clearly progresses from the dayside to the nightside. This suggests that even if low-level nightside reconnection does occur, it doesn't affect the observed convection pattern.

In our simulations, we are very certain that tail reconnection and its outflow are not active during the interval under study. We have included additional discussions on this point in Line 329-338.

Cowley, S. W. H. & Lockwood, M. Excitation and decay of solar wind-driven flows in the magnetosphere-ionosphere system. *Annales Geophysicae* 10, 103–115 (1992).

Grocott, A., Cowley, S. W. H., Sigwarth, J. B., Watermann, J. F. & Yeoman, T. K. Excitation of twin-vortex flow in the nightside high-latitude ionosphere during an isolated substorm, *Annales Geophysicae* 20: 1577–1601, (2002).

Ridley, A. J., Gombosi, T. I. & DeZeeuw, D. L. Ionospheric control of the magnetosphere: conductance, *Annales Geophysicae* 22: 567–584 (2004).

Milan, S. E. Modeling Birkeland currents in the expanding/contracting polar cap paradigm, *J. Geophys. Res. Space Physics*, 118, 5532–5542, doi:10.1002/jgra.50393 (2013).

Milan, S. E. & Grocott, A. High latitude ionospheric convection, in *Space Physics and Aeronomy Collection Volume 3: Ionosphere Dynamics and Applications*, Geophysical Monograph 260, eds. C. Huang 10 and G. Lu, American Geophysical Union, Wiley and Sons, 21-47, <https://doi.org/10.1002/9781119815617.ch2> (2021).

REVIEWER COMMENTS

Reviewer #1 (Remarks to the Author):

The manuscript has been significantly improved since the last round of reviews. I have a few small comments, so I recommend the manuscript for minor revisions. Thank you for inviting me to review this paper, and apologies for the delay in returning my review.

Comments on responses

- In data section, ref 43 is cited for the process of producing SuperDARN convection maps. The original reference, which must be included here as it is the original, is: Ruohoniemi, J. M. and Baker, K. B. Large-scale imaging of high-latitude convection with Super Dual Auroral Radar Network HF radar observations. *Journal of Geophysical Research: Space Physics*, 103(A9):20797-20811, 1998. doi: 10.1029/98JA01288. (your reference 27)
- Line 133-134: "Defining FAC at the top of the ionosphere is essentially equivalent to determining ionospheric convection for a given conductivity" I am not sure what this statement means exactly and if it really makes sense. In theory, a measurement of FACs should give you an indication of what the convection is doing, but it is certainly not a one-to-one correspondence. Please remove this statement. Additionally, I do not feel you have described the canonical convection pattern – a simple sentence will do (with citation) to say something like: "under Dungey-driven convection we see a roughly two cell convection pattern with antisunward flow across the polar cap and sunward flow at lower latitudes."
- On my comment "Introduction, paragraph 3, "approximately 10 minutes" is this to suggest that the entire cycle of convection, tracing a particular field line, occurs in 10 minutes? – this is remarkably quick given how slow the ionosphere is to respond to large scale changes." Your response to this in the rebuttal document is very clear, but I don't feel you have explained it as well in the text of the paper. Please could you expand the text in the paper to the same standard as the rebuttal document.
- On my comment "Fig 2(a3) vx is clearly the dominant component of velocity in this interval. Because of the difference in magnitude between vx and other components, you can't really see the variation on vx. I suggest presenting just vx with an appropriate yscale so that you can see the variation (or indeed, lack of)." I appreciate that it is also important to consider vy, and the amendments you have made. I still think it is tricky to see the evolution of vx and other components due to differences in magnitude. Perhaps you could present one panel just vx and another panel with vy and vz. If they are important, we must be able to see their evolution!
- On my comment "Fig 2 b2-e2, I can only see two lines mapping invariant latitude, but the caption says there are three." I appreciate your explanation but do not feel you have addressed my comment. Perhaps you could make the lines thicker, so they are more clear for the reader?
- Note to the editor on my comment "Section 2.1, final paragraph, "private communications" note to editor – are private communications allowed?"

Other comments

- Line 223 "based on magnetic field data from Cluster 4 near Earth's bow shock". This statement is a bit vague, how exactly did you do this? Was it from visual inspection or automatically?

Reviewer #2 (Remarks to the Author):

I am glad to see that the authors have taken my suggestions seriously and conducted the suggested numerical experiments. As a result the title and a large part of the article have changed. I believe the results presented in the revised version are better supported by the numerical evidence. The revised version may require some proofreading for grammar and typos, but otherwise it is fine. The authors have addressed my concerns.

Reviewer #3 (Remarks to the Author):

Review of revised manuscript: Global-scale Magnetosphere Convection Driven by Dai et al.

The manuscript has been substantially revised from the initially submitted version. In particular, it no longer claims to be discussing a new concept whereby magnetospheric convection can be driven by dayside reconnection (the "Expanding" part of the Expanding-Contracting Polar Cap (ECPC) paradigm), but instead focusses on new model data that supports this rather well-established concept. I am still a little unsure just how novel the findings of the paper are, given the ~30 years of work in this area since the concept was first proposed, but it is interesting (and I think not common) to see this particular aspect of the ECPC model being demonstrated. I therefore think it will make a worthwhile addition to the literature, but does first still need a few issues addressing:

Line 147: It is stated that "ECPC models do not directly address magnetospheric convection; rather, the models imply it through ionospheric convection". This is not true. The ECPC model is at its core a theoretical model, that discusses the coupled magnetosphere-ionosphere system. It is not meaningful to suggest the model only address ionospheric convection. In the early Cowley and Lockwood (1992) paper, for example, both magnetospheric and ionospheric effects of the ECPC model are considered. It is true that the model is often interpreted in terms of its ionospheric consequences, largely due to the difficulties in observing the global magnetosphere that are noted in the present paper.

It is further suggested that a strength of this paper (over previous ionospheric-based ECPC studies) is that it does not need to rely on "mapping convection electric fields from the ionosphere to the magnetosphere" (line 151). But on line 205 it is stated that "The magnetosphere-ionosphere coupling follows the scheme in [25, 35, 37], involving a mapping of FAC from the magnetosphere inner boundary to the ionosphere and convection electric field from the ionosphere to the magnetosphere inner boundary." So, it appears that even in the present paper, the mapping between the magnetosphere and ionosphere is used to interpret the data for this event. If that is not quite true then this needs explaining / rephrasing.

Line 250: It is stated that "The latest version of the radar software toolkit (RST 4.3) of SuperDARN is utilized to produce the potential map every two minutes." This information is not sufficient for a reader to reproduce the analysis. There are various steps involved in the radar data processing that will impact the resulting maps. I suggest either including this info in an appendix or perhaps in the zenodo record would be sufficient for the journal?

387: In Fig. 4 j-k I would also say that the weaker enhancement in the flows at midnight of 500-600 m/s are also indicative of substorm expansion phase flows.

509: convention > convection

457: It is stated that "the impact on convection is instantaneous from the dayside to the nightside for a specified dayside reconnection in the ECPC model." This is not true. As explained by Cowley and Lockwood (1992) the expected timescale of flow excitation across the whole polar cap is 15 mins. This is wholly consistent with the 10-20 min timescale found in the present paper.

451: "The ECPC models compute ionosphere convection based on assumed parameters of the dayside reconnection driver at a specific time. Therefore, examining the temporal evolution of convection/FAC in ECPC depends on presumed knowledge of the temporal evolution of dayside reconnection." The ECPC model is a theoretical model and as such does not compute anything based on assumptions of anything else. It is true that many studies employing the ECPC model do use estimated/proxy measurements of the dayside reconnection rate, and that is a limitation of those studies, but not a limitation of the ECPC model.

References:

Cowley, S. W. H. & Lockwood, M. Excitation and decay of solar wind-driven flows in the magnetosphere-ionosphere system. *Annales Geophysicae* 10, 103-115 (1992).

We sincerely appreciate the feedback from all reviewers. These comments have been highly valuable in guiding our studies. Below, we provide detailed and one-to-one responses.

REVIEWER COMMENTS

Reviewer #1 (Remarks to the Author):

The manuscript has been significantly improved since the last round of reviews. I have a few small comments, so I recommend the manuscript for minor revisions. Thank you for inviting me to review this paper, and apologies for the delay in returning my review.

Comments on responses

- In data section, ref 43 is cited for the process of producing SuperDARN convection maps. The original reference, which must be included here as it is the original, is: Ruohoniemi, J. M. and Baker, K. B. Large-scale imaging of high-latitude convection with Super Dual Auroral Radar Network HF radar observations. *Journal of Geophysical Research: Space Physics*, 103(A9):20797-20811, 1998. doi: 10.1029/98JA01288. (your reference 27)

Thank you for this comment. We have now included the original reference for producing SuperDARN convection maps.

- Line 133-134: “Defining FAC at the top of the ionosphere is essentially equivalent to determining ionospheric convection for a given conductivity” I am not sure what this statement means exactly and if it really makes sense. In theory, a measurement of FACs should give you an indication of what the convection is doing, but it is certainly not a one-to-one correspondence. Please remove this statement. Additionally, I do not feel you have described the canonical convection pattern – a simple sentence will do (with citation) to say something like: “under Dungey-driven convection we see a roughly two cell convection pattern with antisunward flow across the polar cap and sunward flow at lower latitudes.”

Thank you for this comment. We have removed the ambiguous statement.

In addition, we have implemented your suggestion to add more descriptions on the Dungey convection pattern. “Correspondingly, the Dungey convection in the ionosphere is roughly a two cell convection pattern, characterized by antisunward flow across the polar cap and sunward flow at lower latitudes [Dungey1961].” Specific modifications can be found in Line 85-88.

- On my comment “Introduction, paragraph 3, “approximately 10 minutes” is this to suggest that the entire cycle of convection, tracing a particular field line, occurs in 10 minutes? – this is remarkably quick given how slow the ionosphere is to respond to large scale changes.” Your response to this in the rebuttal document is very clear, but

I don't feel you have explained it as well in the text of the paper. Please could you expand the text in the paper to the same standard as the rebuttal document.

Thank you for your comment. We agree that this adjustment should have been made in the previous revision.

In this current revision, we have provided additional details on this point. "The chain of dayside-driven convection is expected to occur within a short timescale (10-20 minutes) of re-establishing the ionosphere's two-cell convection. This timescale reflects the rapid phase propagation of an 'enhanced convection' state across the global magnetosphere and ionosphere. Within this 10-20 minute window, the plasma associated with a specific field line moves only a relatively short distance throughout the entire convection cycle, which typically lasts 2-4 hours [Kennel, 1996]". Specific modifications can be found in Line 103-113.

- On my comment "Fig 2(a3) vx is clearly the dominant component of velocity in this interval. Because of the difference in magnitude between vx and other components, you can't really see the variation on vx. I suggest presenting just vx with an appropriate yscale so that you can see the variation (or indeed, lack of)." I appreciate that it is also important to consider vy, and the amendments you have made. I still think it is tricky to see the evolution of vx and other components due to differences in magnitude. Perhaps you could present one panel just vx and another panel with vy and vz. If they are important, we must be able to see their evolution!

Thank you for your comment. Following your suggestion, we have included the vx panel in Figure 2. The y-scale of vx has been modified to [-150 km/s, 150 km/s] for better visualization of its evolution. In Supplementary Figure S2, we adjust the y-scale of vy to -100 km/s to 100 km/s. Notably, the contribution from vy is observed in its quadrupole distribution within the 65°-75° range at T=7-9 min. The specific modifications can be found in Line 284-287.

- On my comment "Fig 2 b2-e2, I can only see two lines mapping invariant latitude, but the caption says there are three." I appreciate your explanation but do not feel you have addressed my comment. Perhaps you could make the lines thicker, so they are more clear for the reader?

Thank you for this comment. We have addressed this by making corrections to increase the thickness of the lines.

- Note to the editor on my comment "Section 2.1, final paragraph, "private communications" note to editor – are private communications allowed?"

Thank you for this comment. We have addressed this by removing the notes on private communications.

Other comments

- Line 223 "based on magnetic field data from Cluster 4 near Earth's bow shock". This statement is a bit vague, how exactly did you do this? Was it from visual inspection or

automatically?

Thank you for your comment. The analysis was conducted through visual inspection. In the revision, we have incorporated magnetic field data from Cluster 4 for this event in the supplementary figure. The clear signal of southward turning of Bz near 21:43 UT is observable in Cluster. Specific modifications are in Line 225-226.

Reviewer #2 (Remarks to the Author):

I am glad to see that the authors have taken my suggestions seriously and conducted the suggested numerical experiments. As a result the title and a large part of the article have changed. I believe the results presented in the revised version are better supported by the numerical evidence. The revised version may require some proofreading for grammar and typos, but otherwise it is fine. The authors have addressed my concerns.

We greatly appreciate your comments and evaluation!

Reviewer #3 (Remarks to the Author):

Review of revised manuscript: Global-scale Magnetosphere Convection Driven by Dai et al.

The manuscript has been substantially revised from the initially submitted version. In particular, it no longer claims to be discussing a new concept whereby magnetospheric convection can be driven by dayside reconnection (the “Expanding” part of the Expanding-Contracting Polar Cap (ECPC) paradigm), but instead focusses on new model data that supports this rather well-established concept. I am still a little unsure just how novel the findings of the paper are, given the ~30 years of work in this area since the concept was first proposed, but it is interesting (and I think not common) to see this particular aspect of the ECPC model being demonstrated. I therefore think it will make a worthwhile addition to the literature, but does first still need a few issues addressing:

Line 147: It is stated that “ECPC models do not directly address magnetospheric convection; rather, the models imply it through ionospheric convection”. This is not true. The ECPC model is at its core a theoretical model, that discusses the coupled magnetosphere-ionosphere system. It is not meaningful to suggest the model only address ionospheric convection. In the early Cowley and Lockwood (1992) paper, for example, both magnetospheric and ionospheric effects of the ECPC model are considered. It is true that the model is often interpreted in terms of its ionospheric consequences, largely due to the difficulties in observing the global magnetosphere that are noted in the present paper.

It is further suggested that a strength of this paper (over previous ionospheric-based ECPC studies) is that it does not need to rely on “mapping convection electric fields from the ionosphere to the magnetosphere” (line 151). But on line 205 it is stated that “The magnetosphere-ionosphere coupling follows the scheme in [25, 35, 37], involving a mapping of FAC from the magnetosphere inner boundary to the ionosphere and convection electric field from the ionosphere to the magnetosphere inner boundary.” So, it appears that even in the present paper, the mapping between the magnetosphere and ionosphere is used to interpret the data for this event. If that is not quite true then this needs explaining / rephrasing.

Thank you for your comment. We agree that Line 147 and Line 151 need to be rephrased for improved accuracy.

The general ECPC model addresses the interplay of polar cap motion, aurora motion, magnetospheric-ionospheric convection, and dayside/night reconnection. In this broad sense, our study represents a numerical version of the ECPC model, specifically focusing on dayside reconnection and magnetospheric convection with actual data.

Considering the above, we have revised the paragraph containing Line 147 and Line 151 as follows: "Firstly, previous studies of ECPC model often focus on ionospheric convection data. Our approach involves the examination of global simulation and observational data of magnetospheric convection. The combination of global simulations with magnetospheric observations can provide direct evidence of dayside-driven magnetospheric convection. Secondly, studies involving simplified ECPC models typically specify the dayside reconnection driver to compute ionospheric convection for mathematical simplicity [Siscoe and Huang 1985, Milan2013]. In contrast, our global simulations analyze the temporal evolution and the self-consistent interplay of magnetic reconnection, magnetospheric convection, and ionospheric convection. For example, the ionospheric convection electric fields and magnetosphere FAC resulting from dayside reconnection are self-consistently computed in a loop [Raeder2003]. Our study is effectively a numerical version of ECPC model that directly address dayside-driven magnetospheric convection with actual data." Specific modifications are in Line 147-173.

Line 250: It is stated that “The latest version of the radar software toolkit (RST 4.3) of SuperDARN is utilized to produce the potential map every two minutes.” This information is not sufficient for a reader to reproduce the analysis. There are various steps involved in the radar data processing that will impact the resulting maps. I suggest either including this info in an appendix or perhaps in the zenodo record would be sufficient for the journal?

Thank you for this comment. We move this information to the appendix.

387: In Fig. 4 j-k I would also say that the weaker enhancement in the flows at midnight of 500-600 m/s are also indicative of substorm expansion phase flows.

Thank you for this comment. We add this information in the revision. Specific modifications are in Line 389-390.

509: convention > convection

Thank you. We have made the correction.

457: It is stated that “the impact on convection is instantaneous from the dayside to the nightside for a specified dayside reconnection in the ECPC model.” This is not true. As explained by Cowley and Lockwood (1992) the expected timescale of flow excitation across the whole polar cap is 15 mins. This is wholly consistent with the 10-20 min timescale found in the present paper.

Thank you for your comment. We have incorporated this point into the revision. "The 10-20 minute timescale of progression is consistent with the estimated timescale of flow excitation across the whole polar cap in the ECPC model [Cowley and Lockwood, 1992]." Modifications are in Line 452-456.

451: “The ECPC models compute ionosphere convection based on assumed parameters of the dayside reconnection driver at a specific time. Therefore, examining the temporal evolution of convection/FAC in ECPC depends on presumed knowledge of the temporal evolution of dayside reconnection.” The ECPC model is a theoretical model and as such does not compute anything based on assumptions of anything else. It is true that many studies employing the ECPC model do use estimated/proxy measurements of the dayside reconnection rate, and that is a limitation of those studies, but not a limitation of the ECPC model.

We agree with this perspective. We have rephrased these statement accordingly.

“Our global simulations reveal a 10-20 minute progression of magnetosphere convection and FAC from the dayside towards the nightside. This result is achieved through self-consistent computation of the temporal evolution of magnetosphere convection, Region 1/2 FAC, and dayside reconnection. The 10-20 minutes timescale of progression is consistent with the estimated timescale of flow excitation across the whole polar cap in ECPC model [Cowley and Lockwood 1992].” Modifications are in Line 449-456.

References:

Cowley, S. W. H. & Lockwood, M. Excitation and decay of solar wind-driven flows in the magnetosphere-ionosphere system. *Annales Geophysicae* 10, 103–115 (1992).

REVIEWER COMMENTS

Reviewer #1 (Remarks to the Author):

The authors have made the corrections I have suggested and I feel my concerns have been addressed. I am happy with the current draft. I recommend this for publication!

Reviewer #3 (Remarks to the Author):

The authors have considered all my comments and I think the paper is now much more suitable for publication. A couple of remaining queries:

1. The authors state that they have provided full details of the SuperDARN convection mapping procedure used in an Appendix. But I do not see an appendix. In the "Data set from Observations" section the authors refer only in a general way to how the processing is conducted, and this is insufficient for a reader to replicate the results.
2. I noticed a comment from the other reviewer about the V_x data in Fig. 2a3, that a single vertical axis range was inappropriate for the data (3 V components) due to the very different magnitudes of the components (V_x in particular). I agree with this comment. The authors state that they changed the range for V_x in its own panel, but I see the same old version of the figure in the manuscript.

We sincerely appreciate the valuable feedback and evaluations from all the reviewers. We believe that the manuscript has significantly improved in response to the constructive feedback received.

REVIEWER COMMENTS

Reviewer #1 (Remarks to the Author):

The authors have made the corrections I have suggested and I feel my concerns have been addressed. I am happy with the current draft. I recommend this for publication! We greatly appreciate your comments and evaluation!

Reviewer #3 (Remarks to the Author):

The authors have considered all my comments and I think the paper is now much more suitable for publication. A couple of remaining queries:

1. The authors state that they have provided full details of the SuperDARN convection mapping procedure used in an Appendix. But I do not see an appendix. In the "Data set from Observations" section the authors refer only in a general way to how the processing is conducted, and this is insufficient for a reader to replicate the results.

Thank you for your feedback. Following your suggestion, we have provided detailed information on implementing the Radar Software Toolkit (RST) to generate SuperDARN convection maps in the Zenodo record, as recommended in the previous report. This information is referred in the "Data Availability" section.

“In the first step, we process SuperDARN RAWACF files from all available radars in the Northern Hemisphere using the `make_fit` routine. The resulting `fitacf` files contain key physical properties of SuperDARN backscatter, including power, velocity, spectral width, elevation, and associated errors. To enhance the quality of the `fitacf` files, we employ the `fit_speck_removal` routine, which effectively eliminates noise and despecks the files.

In the second step, we generate grid files from the `fitacf` files. These grid files comprise geo-magnetically located line-of-sight velocity vectors. The `make_grid` routine is employed with default options to create grid files for all available radars in the northern hemisphere during the specified event intervals. Subsequently, we merge these individual grid files into a consolidated file using the `combine_grid` routine.

In the third step, we utilize the MAP POTENTIAL packet to generate the convection map from the grid file. The `map_grd` routine is used to reformat the grid file into a `cnvmap` format file, using the default option of AACGM-v2. The `map_addimf` routine

is used to add solar wind OMIN data to the map file. Additionally, the map_addmodel routine calculates the TS18 statistical model (Thomas and Shepherd, 2018), incorporating it into the convection map file with default options. Finally, the map_fit routine, with default options, performs spherical harmonic fitting to generate the final convection map file.

Reference:

Thomas, E. G., and S. G. Shepherd (2018), Statistical patterns of ionospheric convection derived from mid-latitude, high-latitude, and polar SuperDARN HF radar observations, *J. Geophys. Res. Space Physics*, 123, 3196-3216, doi:10.1002/2018JA025280. “

2. I noticed a comment from the other reviewer about the Vx data in Fig. 2a3, that a single vertical axis range was inappropriate for the data (3 V components) due to the very different magnitudes of the components (Vx in particular). I agree with this comment. The authors state that they changed the range for Vx in its own panel, but I see the same old version of the figure in the manuscript.

Thank you for your comment. We have implemented the changes recommended by Reviewer 1, specifically re-adjusting the range for Vx and Vy.

The adjustment made may seem very small in panels at T=3min and T=5min due to the relatively low strength of magnetosphere convection during these intervals. However, as magnetosphere convection intensifies at T=7min and T=9min, the panels of Vx with the re-adjusted y-scale (-150km to 150kms) exhibit more noticeable differences compared to the previous version with the y-scale [-200km to 200kms].

REVIEWERS' COMMENTS

Reviewer #3 (Remarks to the Author):

The authors have now addressed one of my points (about the SuperDARN analysis procedure) - thank you.

Regarding the other point, about Figure 2a3, however, the figure still seems not to have been changed - in the latest review response the authors seem perhaps to be referring to a different figure? I have attached a screen grab of Figure 2a3 for clarity. The vx variation is almost imperceptible owing to it being on the same y-scale as the other components. In their earlier response the authors said that a new version of the figure with a separate vx panel was included, but it still seems not to be.

Reviewer #3 (Remarks to the Author):

The authors have now addressed one of my points (about the SuperDARN analysis procedure) - thank you.

Regarding the other point, about Figure 2a3, however, the figure still seems not to have been changed - in the latest review response the authors seem perhaps to be referring to a different figure? I have attached a screen grab of Figure 2a3 for clarity. The vx variation is almost imperceptible owing to it being on the same y-scale as the other components. In their earlier response the authors said that a new version of the figure with a separate vx panel was included, but it still seems not to be.

I apologize for the oversight in my previous responses regarding Fig. 2a3. I appreciate your clarification and careful review. We realize that the discussions are about Vx in Fig. 2a3 rather than Fig. 2d.

In this revised version, we have accurately addressed the concerns related to Fig. 2a. Specifically, Vx in the previous Fig. 2a3 is now presented with a y-scale ranging from -350 km/s to -450 km/s. Additionally, Vy and Vz are now depicted in Fig. 2a4 with a y-scale spanning from -100 km/s to 100 km/s. The requested modifications for Vx, Vy, and Vz have been incorporated into the updated panels provided below.